# The Comparative Effect of Reduced Mindfulness-Based Stress on Heart Rate Variability among Patients with Breast Cancer

**DOI:** 10.3390/ijerph19116537

**Published:** 2022-05-27

**Authors:** Shu-Jung Wang, Yun-Chen Chang, Wen-Yu Hu, Yuh-Ming Chang, Chi Lo

**Affiliations:** 1School of Nursing, College of Medicine, National Taiwan University, No. 1, Ren-Ai Rd. Sec. 1, Taipei 10051, Taiwan; wsj2785@gmail.com (S.-J.W.); weyuhu@ntu.edu.tw (W.-Y.H.); 2School of Nursing and Graduate Institute of Nursing, China Medical University, No. 100, Sec. 1, Jingmao Rd., Beitun Dist., Taichung 40604, Taiwan; 3Department of Nursing, National Taiwan University Hospital, No. 7, Chung-Shan South Rd., Taipei 10002, Taiwan; 4Institute of Biochemistry, Microbiology, and Immunology, Chung Shan Medical University, Taichung 40201, Taiwan; ymc02468@gmail.com; 5Department of Hospitality Management, Chung Hua University, Hsinchu 30012, Taiwan; chilo@chu.edu.tw

**Keywords:** heart rate variability (HRV), breast cancer, mindfulness-based stress reduction (MBSR)

## Abstract

Heart rate variability (HRV) is a powerful tool for observing interactions between the sympathetic and parasympathetic nervous systems. This study evaluated HRV during a mindfulness-based stress reduction (MBSR) program among women with breast cancer after receiving treatment. A quasi-experimental, nonrandomized design was used. Patients were allocated to usual care (*n* = 25) and MBSR (*n* = 25) groups. HRV was measured using recognized methods to assess the autonomic nervous system. Two-way ANOVA and *t*-tests were used to examine HRV changes between and within groups, respectively. A significant interaction effect of time with group was observed on heart rate (F (1, 96) = 4.92, *p* = 0.029, η^2^ = 0.049). A significant difference was also observed within the MBSR group preintervention and postintervention with regard to heart rate (t (24) = −3.80, *p* = 0.001), standard deviation of the RR interval (t (24) = 5.40, *p* < 0.001), root-mean-square difference in the RR interval (t (24) = 2.23, *p* = 0.035), and high-frequency power (t (24) = 7.73, *p* < 0.001). Large effect sizes for heart rate and SDNN of 0.94 and 0.85, respectively, were observed between the MBSR and usual care groups. This study provides preliminary evidence that an MBSR program may be clinically useful for facilitating parasympathetic activity associated with feelings of relaxation in treated breast cancer survivors.

## 1. Introduction

In Taiwan, breast cancer is the most common cancer among women and is the second-highest cause of cancer-related death [1]. In the United States, breast cancer death rates in women are higher than those for any other cancer apart from lung cancer [2]. However, an international study found that women with breast cancer who received conventional western medicine (i.e., chemotherapy, hormone therapy) may have adverse effects, compared with healthy women. For example, patients with breast cancer were at increased risk of anxiety, depression, suicide, sexual dysfunction, and neurocognitive effects [3].

The autonomic nervous system (ANS) belongs to the peripheral nervous system and consists of the sympathetic nervous system (SNS) and the parasympathetic nervous system (PNS). They are associated with anxiety and depression and are considered key predictors of cardiovascular events and morbidity [4].

Heart rate variability (HRV) is a measure of beat-to-beat variability in heart rate and provides a noninvasive physiological tool for the observation of the autonomic nervous system (ANS) [5,6,7]. The standard parameters of HRV are related to the ANS function—parameters such as linear (time and frequency domain) and nonlinear (symbolic dynamics and compression entropy) information [8]. Depending on these parameters, sympathetic and parasympathetic modulation can be derived; for instance, parasympathetic activity is associated with RMSSD, HF, and SDNN, while LF and LF/HF are associated with both [9].

In one study, HRV was compared between two groups—breast cancer survivors who had aggressive treatments in the first year vs. healthy women—and cardiovascular imbalances were observed in breast cancer survivors with significantly higher resting heart rates and significantly lower standard deviations of normal-to-normal intervals (SDNN ms), HRV index, as well as with high frequencies (HF ms^2^) of HRV [10]. On the other hand, a low rate of high-frequency HRV is a risk factor for shorter life expectancy in women diagnosed with metastatic or recurrent breast cancer [11]. Breast cancer survivors with a history of distress disorders may have lower autonomic flexibility when exposed to stress. In other words, this finding underscores that a history of distress disorders is a unique correlator of poorer cardiovascular function in breast cancer survivors [12]. The above finding indicates a relationship between autonomic nervous system dysfunction and breast cancer prognosis.

Mindfulness-based stress reduction (MBSR) was developed in 1979 by Jon Kabat-Zinn in the outpatient stress reduction clinic at the University of Massachusetts Medical Center [13]. The MBSR program has been used as an effective stress-reducing intervention and as a model in other healthcare settings [14]. MBSR is an effective psychoeducational program that is recommended for breast-cancer-related complications and has been indicated to improve physical and psychological symptoms in patients with breast cancer [15,16]. A recent study by Lee et al. (2022) found that a mindfulness stress intervention was effective in reducing fear of recurrent symptoms and improving the quality of life among female cancer survivors [17]. In a previous study, 6 weeks of routine meditation practice was associated with changes in the HRV index, and the HRV records of 85% of patients indicated improved homeostatic regulation in the ANS [18]. Arab et al. (2016) demonstrated that HRV analysis could be used as an auxiliary, noninvasive tool for the early diagnosis of autonomic dysfunction to improve the prognosis of patients with breast cancer [19]. Nijjar et al. (2014) enrolled 22 healthy participants and reported that their cardiac sympathovagal balance, measured by HRV, improved after receiving an 8-week MBSR intervention [20].

The outcomes of MBSR for patients with breast cancer are typically measured using self-reported questionnaires to determine whether the patients had physical, psychological, or social problems. However, the measured improvements were insufficiently accurate and could not serve as objective measures of improvement. HRV was used as a metric to quantitatively understand the effects of MBSR on the ANS.

Previous studies have investigated the effects of MBSR intervention on patients with cancer or other illness. In this study, objective parameters were established, and the findings of previous studies were extended using physiological biofeedback measurements. We hypothesized that heart rate, SDNN (ms), RMSSD (ms), low-frequency power (ms^2^), high-frequency power (ms^2^), and LF/HF ratio components of HRV would improve in patients with breast cancer during the MBSR program.

## 2. Materials and Methods

### 2.1. Study Design

A quasi-experimental, nonrandomized study was performed, in which pre–postmonitoring measures were used. Participants were screened and completed a baseline assessment before being randomized to usual care (UC) and MBSR groups. The ethics committee of the hospital approved the study, and all participants provided informed consent.

### 2.2. Sample and Setting

This study used nonprobability sampling types of snowball sampling and convenience sampling. The participants (*n* = 50) were patients with breast cancer who were recruited through telephone invitations, internet advertisements, posts on online forums, and referrals from physicians at the Northern Regional Hospital in Taiwan. The inclusion criteria were diagnosis and treatment for breast cancer within the last 2 years, an age of 20–65 years, and the ability to communicate in Mandarin. Exclusion criteria were breast cancer survivors with cardiovascular disease (CVD) or severe mental illness. No fee was charged for the MBSR sessions. All research procedures were performed in a yoga classroom at a university in northern Taiwan.

### 2.3. Procedures

Participants provided informed consent. The MBSR program was originally an 8-week training course comprising 8 sessions of 2.5 h each [21]. For patient convenience and to increase participation rates, many recent studies have shortened the duration of the program to 6 weeks [15,22,23,24]. Therefore, a 6-week formal session with condensed 2 h sessions suitable for patients with breast cancer was used. Patients should maintain an objective, permissive, nonjudgmental, and open-hearted attitude when perceiving breathing and bodily sensations. Through the sessions, patients with breast cancer attempted to reduce stress to improve their HRV.

### 2.4. Data Collection

All participants submitted written informed consent during the orientation period. Making a variable measurement involves two time points—a pretest and a posttest. (1) Pretest: This time point was no later than the second week of the course; (2) posttest: This time point was to complete the questionnaire within one week after the sixth week of the program. Participants in the experimental group received additional yoga mats. Participants who completed the questionnaire could receive rewards (e.g., coupons).

### 2.5. Sample Size Calculation

The study sample size was calculated for 80% power and 5% type-I error to detect an effect size of 0.5 using a repeated-measures within-between interaction with 2 groups and 2 points of measurement. Thus, a sample size of 34 patients was required. Assuming an attrition rate of 20% at follow-up, we targeted a sample of 40 patients for the intervention and control groups at a two-tailed level of significance of 0.05.

### 2.6. Heart Rate Variability Measurement

HRV is a complementary, noninvasive measure of ANS function [19]. We used a wristband heart rate monitor (ZoeTekTechnology Co., Ltd., Taipei, Taiwan) [25] based on a photoplethysmogram (PPG), which is a professional-grade HRV sensing instrument with several sensors; the device has good test–retest reliability [26,27,28]. HRV analysis was processed using methods previously described in detail [27]. Participants refrained from consuming caffeine and tobacco for 4 h before participating in the MBSR program. After the clinical demographic questionnaire was completed, physiological measurements were taken. The participants sat on comfortable chairs, and the wearable device’s physiological sensor was set up. The recording was performed in a quiet, temperature-controlled room at 25 °C to accurately assess HRV. The electrodes were placed on the wrist, and the signals were recorded for 5 min. All HRV data were collected and transmitted to an Android smartphone via Bluetooth 4.0 [27]. Heart rhythm was recorded, and HRV parameters were expressed as standard time-domain measurements. These parameters were average heart rate, SDNN, root-mean-square differences for successive RR intervals (RMSSD, ms), and frequency-domain measurements—namely, low-frequency power (LF, ms^2^), high-frequency power (HF, ms^2^), and the ratio of absolute LF power to HF power (LF/HF).

### 2.7. MBSR Intervention

The MBSR intervention comprised two groups, with 6 weeks of 2 h sessions held once a week. Participants received a formal MBSR program [21] comprising 5 key forms of training: (1) Mindful eating techniques (3 raisins were given to each person; they were then asked to appreciate the food given and to achieve an integrated awareness of all their senses while eating); (2) body scan (participants were asked to lie or sit comfortably and focus on the sensation of their bodies from head to toe); (3) sitting meditation (awareness of one’s breath, bodily sensations, and thoughts, and ambient sound); (4) gentle and friendly Hatha yoga (participants’ current physical state determined to what extent they could stretch, and participants were aware of changes in their body at all times; natural breathing was used); and (5) walking meditation (awareness of bodily changes while walking). Patients were asked to perform daily informal mindfulness practices with audio played through a CD for 10–15 min/day, 5–6 times/week. They were also required to record their practice time, postpractice awareness, stressful situations, stress responses, etc., on a construct paper log (Table 1).

### 2.8. Fidelity of Intervention

In order to enhance intervention fidelity, we used a previously written protocol for each session, ensuring its application to all participants equally, and invited a qualified mindfulness facilitator with over 9 years of MBSR system training to be in charge of the 6-week intervention. A co-investigator (co-PI) checked patient informal practice records per week during the program. On the other hand, in order to improve fidelity, participants were asked to follow the training manual during the 2 h introduction meeting. Moreover, the first author contacted the participants through the messaging mobile app LINE each week to remind patients to practice and attend the next session. She also motivated participants and answered any questions they might have related to the training. The limitations of fidelity in our study were that there was no other comparable intervention group, nothing was performed with the control group, no follow-up was carried out, and samples from other centers were not used to increase their representativeness.

### 2.9. Usual Care (UC) Group

To control for potential confounders, patients with breast cancer in the UC group did not perform meditation, yoga, or other alternative therapies for 6 weeks; they received routine care. UC group patients completed all postintervention measures, and MBSR group patients completed the treatment. Patients with breast cancer in the UC group could choose to receive MBSR intervention for another 6 consecutive weeks after completing the postintervention measures.

### 2.10. Statistical Methods

Statistical analyses were performed using SPSS 22.0 software (IBM, Chicago, IL, USA). Patients had to have attended at least five sessions to be included in the statistical analysis. The frequencies, percentages, means, and standard deviations of the participants’ characteristics were calculated. Between-group differences were assessed using the chi-square test for categorical variables and independent *t*-tests for continuous variables. The Kolmogorov–Smirnov test was used to determine whether the data were normally distributed. To analyze the effects of the MBSR program, two-way ANOVA was used; the group was the between-group variable, and paired-sample *t*-tests were used. Independent-sample *t*-tests were used for within-group variables. ANOVA effect sizes were calculated using partial eta-squared (η^2^), and Cohen’s d for independent-sample *t*-tests was used for comparison of effect sizes. Effect sizes of d = 0.2 were considered small, 0.5 were medium, and >0.8 were large [29]. Effect sizes smaller than d = 0.2 were considered to indicate no effect, 0.01 ≤ ηp^2^ < 0.058 indicated a small effect, 0.058 ≤ ηp^2^ < 0.138 indicated a medium effect, and ηp^2^ ≥ 0.138 indicated a large effect [29].

## 3. Results

### 3.1. Participant Recruitment and Characteristics

The participant recruitment process is presented as a flowchart in Figure 1. The demographic and clinical characteristics of the participants are displayed in Table 2. Of the 71 eligible participants, 9 women declined participation prior to the baseline assessment. We assigned participants to control group (*n* = 30) and MBSR group (*n* = 32), respectively. The control group and the experimental group dropout rates were five and eight participants, respectively, during the intervention. Fifty women with breast cancer completed the postintervention period; their mean ± SD age was 47.26 ± 8.64, and mean time from diagnosis with breast cancer was 19.36 months. The majority of participants were university education level (*n* = 19, 38%), followed by junior college education (*n* = 12, 24%). With regard to occupation, most were retired (*n* = 12, 24%). Most participants (*n* = 22, 44%) had a diagnosis of stage II breast cancer and were receiving hormonal therapy (*n* = 25, 50%). Most patients were not regular caffeine users (*n* = 31, 62%). No significant differences in baseline demographic characteristics were observed between the MBSR and UC groups, except for smoking history (*p* = 0.04). However, all patients were nonsmokers at the time of the present study.

### 3.2. Intervention Effects

#### 3.2.1. Group Differences at Pretest and Posttest

Statistics including effect sizes and pre- and posttest mean (SD) are presented in Table 3. The MBSR group had a mean pretest heart rate of 88.6 bpm, higher than the posttest heart rate of 82.12 bpm. The mean posttest heart rate of the MBSR group was also lower than that of the UC group (90.16 bpm). We observed a significant interaction effect of heart rate with time by group (F (1, 96) = 4.92, *p* = 0.029, η^2^ = 0.049), a difference in SDNN between groups (*p* = 0.040), and a significant variation in RMSSD over time (before vs. after the training, *p* = 0.024). The two-way ANOVA results revealed a significant reduction in the heart rate of the MBSR group, compared with the UC group (Figure 2).

No other significant between-group effects were observed for the MBSR group and the UC group at the pretest and posttest for SDNN (F (1, 96) = 11.81, *p* = 0.110, η^2^ = 0.110), RMSSD (F (1, 96) = 0.004, *p* = 0.952, η^2^ < 0.001), LF (F (1, 96) = 0.57, *p* = 0.454, η^2^ = 0.006), HF (F (1, 96) = 1.24, *p* = 0.269, η^2^ = 0.013), and HF/LF (F (1, 96) = 0.55, *p* = 0.461, η^2^ = 0.006). Large effect sizes were observed for heart rate and SDNN between the MBSR and the UC groups with effect sizes of 0.94 and 0.85, respectively (Table 3).

#### 3.2.2. Within-Group Effects

The paired-sample *t*-test analyses revealed significant differences within the MBSR group preintervention and postintervention for heart rate (t (24) = −3.80, *p* = 0.001), SDNN (t (24) = 5.40, *p* = 0.000), RMSSD (t (24) = 2.23, *p* = 0.035), and HF (t (24) = 7.73, *p* = 0.000). However, significant differences were also observed within UC group preintervention and postintervention for SDNN (t (24) = −0.58, *p* = 0.031) (Table 3). No statistically significant difference within the MBSR or UC group was observed for LF or LF/HF. The metrics of LF and LF/HF had small effect sizes of 0.21 and 0.23, respectively (Table 3).

## 4. Discussion

Robust, randomized controlled trials (RCTs) for psychological interventions are often more complex than those for drugs, and psychological treatment RCTs are difficult and expensive to perform [30]. Therefore, a nonrandomized, controlled trial design was used in this study to explore the effect of MBSR on HRV in patients with breast cancer.

HRV is a noninvasive objective tool that directly predicts functional health in female breast cancer survivors, which may be applicable in clinical practice [31]. The findings partial strongly supported our study hypotheses that patients participating in MBSR would have improved regulation of time-domain metrics of SDNN (ms), RMSSD (ms), and the frequency-domain metric of heart rate (bpm), HF (ms^2^) in autonomic nervous system activity after received the 6-week MBSR intervention. A large effect size was also observed for SDNN; SDNN was significantly higher in the MBSR group, compared with the UC group. The results of our study are consistent with those of previous studies. A recent review investigated the association between MBSR and HRV, and most studies in that review reported that time-domain-based SDNN and RMSDD and frequency-domain-based HF increased significantly after 8 weeks of mindfulness training [7]. Similarly, another study observed significantly increased SDNN and HF within a group of patients with nonmetastatic breast cancer receiving 4 weeks of MBSR intervention [32]. From a medical perspective, SDNN indicates stress and mortality [33]; lower SDNN values indicate that an individual’s ability to adapt to stress is low and that their degree of stress is high [34]. Decreased values of RMSDD and HF values are also associated with higher morbidity and mortality risk [35]. RMSDD is correlated with HF, and both RMSSD and HF are parameters of parasympathetic nervous system activity [36]. Resting heart rate is also a measure of parasympathetic nervous system activity and autonomic balance [37]. Studies have reported a positive dose–response relationship between resting heart rate and all-cause mortality [37,38]. Lee et al. reported that lower heart rates predicted longer survival in patients with advanced breast cancer [11,39]. Consistent with our study, Matchim et al. treated breast cancer survivors with MBSR and observed reduced heart rates [40]. Younge et al. reported that a web-based mindfulness intervention significantly reduced heart rate in patients with heart disease [41]. In conclusion, our study demonstrated that MBSR could facilitate parasympathetic activity in breast cancer survivors.

Traditional MBSR is an 8-week program originally developed by Kabat-Zinn [14]. Our study employed the abbreviated 6-week MBSR program in consideration of the health status of the breast cancer survivors. Quantitative statistical results were consistent with qualitative interviews. A participant said, “I often felt so nervous and my heart would be beating wildly, but mindfulness practice brought me peace and smooth breathing”. On the other hand, previous RCTs have demonstrated that a 6-week MBSR program was beneficial for health and resulted in significant improvements in psychological symptoms, fear of recurrence, and quality of life in breast cancer survivors compared with UC [42,43]. A recent systematic review reported that truncated MBSR programs (e.g., 4–7 weeks) are as effective as the traditional 8-week MBSR program in reducing anxiety, depression, and stress in healthcare professionals; the programs were also effective in increasing mindfulness and self-compassion [44].

The mechanism of action in MBSR for specific physical or mental health conditions is unclear. Several scholars have stated that the mechanisms of action in MBSR may be general for physical or psychological variables [45]. The theoretical mechanism is proposed to feature the general themes of exposure, acceptance, attention, and awareness [46,47,48,49,50,51,52]. The MBSR program also generates physiological feedback from the central nervous system (CNS) and ANS. Thayer and Lane proposed the neurovisceral integration model, which posits that cognition and emotions are regulated by interactions between CNS and ANS [53]. Structurally, several functionally interconnected regions of the forebrain, limbic system, and brainstem were proposed to coordinate autonomic and behavioral responses to physiological stress [53]. Previous neuroimaging studies have demonstrated that activity in the prefrontal cortex is associated with HRV [54,55]. Neuroimaging studies have also demonstrated that mindfulness training leads to changes in functional connectivity with the prefrontal cortex and linked MBSR to alterations in the ANS and emotional regulatory capabilities [56,57].

The results of this study suggest that MBSR training effectively improves HRV. These findings aid the implementation of interventions to strengthen the parasympathetic nervous system.

### Strengths and Limitations

This study has several strengths. First, it contributes to HRV research by using physiological biofeedback parameters to determine the effects of MBSR intervention in contrast to other studies that have typically only used self-reported questionnaires. Second, few studies have investigated the effects of MBSR intervention on HRV in patients with breast cancer. Our study observed that MBSR had positive effects on heart rate, SDNN, and HF. Third, the patients with breast cancer in our study had excellent adherence to the study protocol, and the sample was homogenous. Finally, our findings indicated that a shorter-than-standard MBSR program may be sufficient to improve ANS function. However, our study also has some limitations. First, the sample size was small; only 50 participants completed the protocol. Second, only the immediate effects of the MBSR program were investigated; whether the positive effects of MBSR persist longitudinally for general health and stress reactivity is unknown. Therefore, our study recommends follow-up of up to 6 months or longer to understand the long-term effect of MBSR intervention in breast cancer survivors. Finally, the COVID−19 pandemic has made group interventions difficult to conduct. Although pandemic prevention measures (i.e., wearing a mask and disinfecting hands with 75% alcohol) were used, participants may have felt psychological stress when gathering with strangers. The effectiveness of internet-based courses could be investigated in future studies.

## 5. Conclusions and Future Works

It is believed that stress promotes the proliferation of cancer cells [58]. The field research in the present study has practical applications. However, implementing these interventions for high-stress patients with breast cancer is challenging. The use of physiological measures of stress to compare the effectiveness of MBSR and UC is a novel contribution of our study. Future studies could use a multicenter design to increase the robustness of the findings and to monitor whether internet-based MBSR and physical forms of MBSR are equally effective in breast cancer patients during the COVID−19 pandemic. Mediation analyses with larger RCT samples can increase the statistical power, and future studies can use more diverse samples with respect to higher incidence cancer type (e.g., colorectal cancer, lung cancer), gender, and culture to generalize this study’s results.

## Figures and Tables

**Figure 1 ijerph-19-06537-f001:**
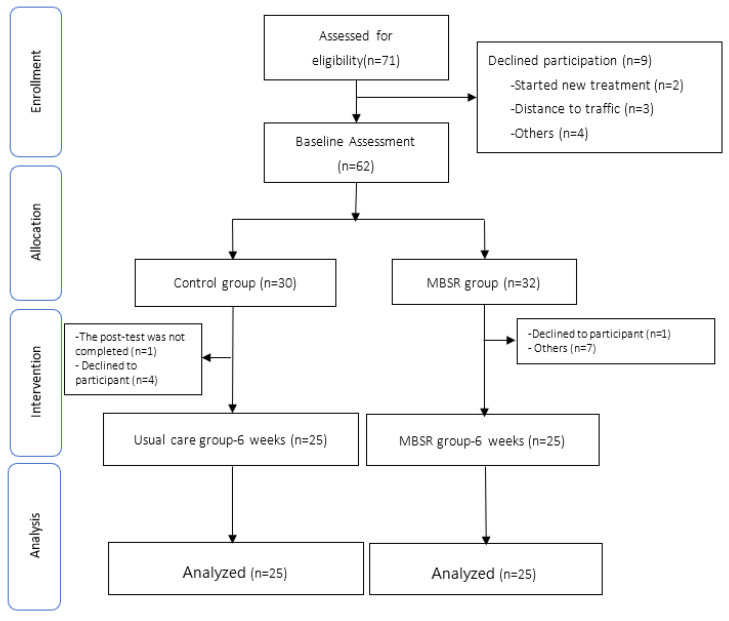
Diagram of participant progression.

**Figure 2 ijerph-19-06537-f002:**
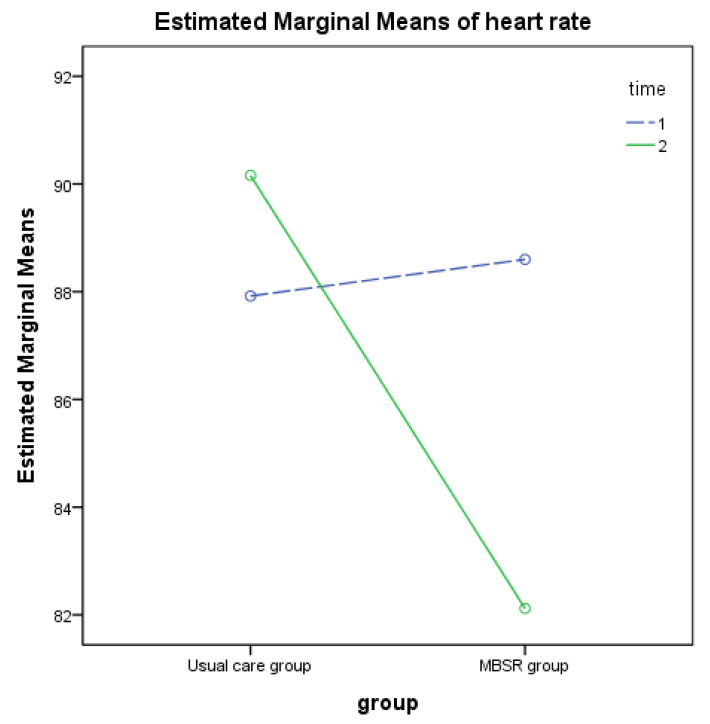
Profile plot showing two-way ANOVA for heart rate on MBSR program.

**Table 1 ijerph-19-06537-t001:** MBSR program in the study.

Week	Themes	Content of the Program	Homework
1	What is mindfulness? Unlock inner wisdom—you have more than you perceive	Mindful eating (raisin tasting) Breath awareness Body scan	Choose a time each day and practice mindful breathing for 5 min. Practice body scans in a sitting or lying position per day. Choose one meal a day and practice mindful eating of at least three mouthfuls.
2	Awareness of body and mind Stress response and coping	Meditation, breathing Three-minute breathing space (S.T.O.*p*.) Mindfulness stretching (standing)	Body scan (sitting or lying), 5–6 times per week. Meditation for 10–15 min, 5–6 times a week. Mindful stretches (joints and glands) per day. Informal practice, recording the most impressive awareness in daily life such as eating, bathing, washing dishes, etc. The inertial response was observed and recorded per day.
3	Mindfulness exercise Balanced breathing	Mindfulness stretching (recumbent position) Alternating nostril breathing	Meditation for 10–15 min, 5–6 times a week. 5–6 times a week, mindful stretching (including joints and glands) once or alternate nostril breathing per day (3 cycles). Observe and record pleasant events per day.
4	Sleep and burnout Ongoing awareness	Deep abdominal breathing Mindful walking	Mindful walk or mindful stretch (joints and glands) per day, 5–6 times a week. Meditation for 10–15 min, 5–6 times a week. Informal practice, recording the most impressive awareness in daily life such as eating, bathing, washing dishes, etc.
5	Facing fear and pain Dealing with Difficulties—our Life Story	Compassionate Choiceless awareness	Mindful walk or mindful stretch (joints and glands) per day, 5–6 times a week. Sit and breathe with Compassionate for 10–15 min, 5–6 times a week. Informal practice, recording the most impressive awareness in daily life such as eating, bathing, washing dishes, etc.
6	Compassionate caring Mindful living	Self-care practice Mountain meditation	Continue any formal practice. Informal practice of incorporating mindfulness into your life.

**Table 2 ijerph-19-06537-t002:** The demographic and clinical characteristic of participants.

Characteristic	Total (*n* = 50) ^a^	MBSR (*n* = 25) ^a^	UC (*n* = 25) ^a^	*p*-Value
Age (years), mean (SD)	47.26 (8.64)	49.28 (10.45)	45.24 (5.90)	0.24
Time since diagnosis (months), mean (SD)	19.36 (15.93)	25.20 (19.42)	13.52 (8.37)	0.10
Educational level, *n*(%)				0.43
Elementary	3 (6.0)	3 (12.0)	0 (0.0)	
Junior	2 (4.0)	0 (0.0)	2 (8.0)	
Senior high	9 (18.0)	2 (8.0)	7 (28.0)	
Junior college	12 (24.0)	6 (24.0)	6 (24.0)	
University	19 (38.0)	12 (48.0)	7 (28.0)	
Graduate institute	5 (10.0)	2 (8.0)	3 (12.0)	
Occupation, *n*(%)				0.77
Retired/none	12 (24.0)	8 (32.0)	4 (16.0)	
Government employees	5 (10.0)	1 (4.0)	4 (16.0)	
Industry/commerce	9 (18.0)	3 (12.0)	6 (24.0)	
Homemaker	8 (16.0)	3 (12.0)	5 (20.0)	
Service industry	7 (14.0)	4 (16.0)	3 (12.0)	
Other	9 (18.0)	6(24.0)	3 (12.0)	
Cancer staging, *n*(%)				0.43
Stage O	4 (8.0)	3 (12.0)	1 (4.0)	
Stage I	8 (16.0)	5 (20.0)	3 (12.0)	
Stage II	22 (44.0)	10 (40.0)	12 (48.0)	
Stage III	7 (14.0)	1 (4.0)	6 (24.0)	
Stage IV	9 (18.0)	6 (24.0)	3 (12.0)	
Treatment, *n*(%)				0.18
Chemotherapy	13 (26.0)	9 (36.0)	4 (16.0)	
Radiotherapy	4 (8.0)	2 (8.0)	2 (8.0)	
Targeted therapy	3 (6.0)	2 (8.0)	1 (4.0)	
Hormone therapy	25 (50.0)	9 (36.0)	16 (64.0)	
Other	5 (10.0)	3 (12.0)	2 (8.0)	
Coffee intake, *n*(%)				0.46
None	31 (62.0)	17 (68.0)	14 (56.0)	
1 cup/day	18 (36.0)	7 (28.0)	11 (44.0)	
2 cups/day	1 (2.0)	1 (4.0)	0 (0.0)	
Tea intake, *n*(%)				0.43
None	26 (52.0)	11 (44.0)	15 (60.0)	
1 cup/day	17 (34.0)	11 (44.0)	6 (24.0)	
2 cups/day	7 (14.0)	3 (12.0)	4 (16.0)	
Smoking history, *n*(%)				0.04 *
No	44 (86.0)	25 (100.0)	19 (76.0)	
Yes	6 (14.0)	0 (0.0)	6 (24.0)	

* *p* ≤ 0.05; SD = standard deviation; ^a^ categorical variable are presented as frequencies and percentages; continuous variables are presented as mean and standard deviation; MBSR = mindfulness-based stress reduction; UC = usual care.

**Table 3 ijerph-19-06537-t003:** Comparisons of HRV parameters between groups over time.

Variable	Group	Pretest	Posttest	Source	F(df1,df2)	*p*	Interaction Time × Group Eta-Squared (η^2^)	Difference (Post–Pre)			
		(Mean ± SD)	(Mean ± SD)					Mean ± SD	t(df)	*p*	Effect Size (d)
Heart rate (bpm)	UC	87.92 ± 8.69	90.16 ± 11.27	G	3.50 (1, 96)	0.064	0.049 *	2.24 ± 9.94	1.13 (24)	0.271	0.94
	MBSR	88.6 ± 10.80	82.12 ± 8.22	T	1.16 (1, 96)	0.284		−6.48 ± 8.52	−3.80 (24)	0.001 *	
				G × T	4.92 (1, 96)	0.029 *					
**Time domain**											
SDNN (ms)	UC	83.34 ± 45.55	72.24 ± 69.47	G	4.04 (1, 96)	0.040 *	0.110	−11.04 ± 95.78	−0.58 (24)	0.031 *	0.85
	MBSR	31.49 ± 15.26	85.81 ± 43.93	T	5.15 (1, 96)	0.051		54.24 ± 50.27	5.40 (24)	0.000 *	
				G × T	11.81 (1, 96)	0.110					
RMSSD (ms)	UC	21.81 ± 8.21	25.96 ± 11.71	G	2.32 (1, 96)	0.131	<0.001 *	6.16 ± 16.89	1.82 (24)	0.081	0.19
	MBSR	19.23 ± 7.20	23.17 ± 7.44	T	5.24 (1, 96)	0.024 *		3.68 ± 8.25	2.23 (24)	0.035 *	
				G × T	0.004 (1, 96)	0.952					
**Frequency Domain**											
LF(ms^2^)	UC	253.58 ± 261.74	616.86 ± 1685.59	G	1.86 (1, 96)	0.176	0.006 *	363.24 ± 1751.17	1.037 (24)	0.310	0.21
	MBSR	146.02 ± 196.06	245.20 ± 371.28	T	1.73 (1, 96)	0.191		99.16 ± 377.40	1.31 (24)	0.201	
				G × T	0.57 (1, 96)	0.454					
HF (ms^2^)	UC	78.96 ± 54.30	222.71 ± 452.27	G	1.15 (1, 96)	0.286	0.013 *	143.6 ± 462.97	1.55 (24)	0.134	0.31
	MBSR	80.75 ± 52.04	121.12 ± 78.06	T	3.93 (1, 96)	0.051		40.4 ± 26.13	7.73 (24)	0.000 *	
				G × T	1.24 (1, 96)	0.269					
LF/HF ratio	UC	4.36 ± 6.91	3.60 ± 2.39	G	4.43 (1, 96)	0.038	0.006 *	−0.76 ± 6.17	−0.62 (24)	0.544	0.23
	MBSR	2.07 ± 2.40	2.50 ± 2.39	T	0.05 (1, 96)	0.831		0.40 ± 2.24	0.89 (24)	0.380	
				G × T	0.55 (1, 96)	0.461					

* *p* ≤ 0.05; bpm = beats per minute; SDNN = standard deviation of all normal-to-normal intervals; RMSSD = root-mean-square of successive differences; LF = low frequency; HF = high frequency; LF/HF = ratio between LF and HF; SD = standard deviation; MBSR = mindfulness-based stress reduction; UC = usual care; G = group; T = time; G × T = group × time.

## Data Availability

The dataset supporting the conclusions of this article is available on request to the corresponding author.

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
