# Peer review of "The Comparative Effect of Reduced Mindfulness-Based Stress on Heart Rate Variability among Patients with Breast Cancer"

_ijerph, 2022, doi:10.3390/ijerph19116537_

Round 1

Reviewer 1 Report

This is an interesting study. The authors have made many improvements suggested by the reviewers and the editor.

Some issues need to be improved:

1) All references must include the doi (https://doi.org/xxxxxx) or the url if it does not exist, which allows the direct location of the publications.

2) References to personal communications should be removed, as they cannot be verified or verified by readers. Authors are sure to find arguments and evidence in relevant international publications that readers can follow.

3) The background should be better clarified and include more current empirical evidence (2022, 2021); as in Discussion and conclusions

4) The background must conclude with the research question, specified in the objective and materialized in the hypothesis

5) They must include a section prior to results explaining the data analysis and modules followed, the arguments used, as well as the software. Anything related to data analysis should be moved to that section and freed from others (results)

6) In results, which must be clarified much better and articulated much better, it must be organized according to the steps indicated and explained before in Data analysis

7) In methodology, the sample, the inclusion and exclusion criteria, representativeness, sampling method, composition x gender x age x characteristics must be clear.

8) In discussion, the answer to the research question must be clear, if the objective is achieved, if the hypotheses are met, the interpretation of what has been provided, the limitations of the study and its way of solving it, etc.

9) Indicators or indices on treatment fidelity should be provided (e.g., previously written protocol and if it was validated or this study is its validation; training of applicators; follow-up of its application; how it was guaranteed that it was applied to all participants in the same way...)

10) The design of the intervention must be clear: number of groups; pre-post-monitoring measures; Session logs and analysis

11) Data and evidence of the construct validity of the measures, convergent validity (extracted mean variance), discriminant validity, composite reliability or MacDonalds omega must be provided. If additional observations or tasks were included, how their inter-observer agreement was checked, how they were recorded, and how their accuracy was ensured

12) All changes must be marked in color for easy checking

Authors are strongly encouraged to make the indicated changes in order to be recommended for publication.

Author Response

Author's Reply to the Review Report (Reviewer 1)

This is an interesting study. The authors have made many improvements suggested by the reviewers and the editor.

Some issues need to be improved:

  • All references must include the doi (https://doi.org/xxxxxx) or the url if it does not exist, which allows the direct location of the publications.

Response:Thank the reviewer for reminding us to present the location of all references available. We have added the DOI of each reference. (page 13, line 389)

  • References to personal communications should be removed, as they cannot be verified or verified by readers. Authors are sure to find arguments and evidence in relevant international publications that readers can follow.

Response:We fully agree with the reviewer’s viewpoint and have replaced the personal communications with systematic review (ref 21).

  • The background should be better clarified and include more current empirical evidence (2022, 2021); as in Discussion and conclusions

Response:thank you very much. We add more current empirical evidence in the Background, Discussion, and Conclusions. (page 2, line 59-62;71-73; page 11, line 272-277; page 12, line 359)

  • The background must conclude with the research question, specified in the objective and materialized in the hypothesis

Response:Thanks for the reviewer’s kind comment. In the last paragraph of the introduction, we have already talked about the research question, purpose and hypothesis. And we marked it in red for easy reading by reviewer. (page 2, line 86-91)

  • They must include a section prior to results explaining the data analysis and modules followed, the arguments used, as well as the software. Anything related to data analysis should be moved to that section and freed from others (results)

Response:Thanks for the reviewer’s kind comment. The section of “2.10. Statistical Methods” was prior to results explaining the data analysis. In the previous revision, we have revised the statistical method according to the editor-in-chief. (page 5, line 188)

  • In results, which must be clarified much better and articulated much better, it must be organized according to the steps indicated and explained before in Data analysis

Response:Thank you for your kind suggestion. We minor modified the results. and check again that its content is correct. (page 5, line 209-212)

  • In methodology, the sample, the inclusion and exclusion criteria, representativeness, sampling method, composition x gender x age x characteristics must be clear.

Response:Thank you so much for this important point. We revised (or added) the statements of sampling method, inclusion and exclusion criteria in the “2.2. Sample and Setting.” (page 3, line 99)

  • In discussion, the answer to the research question must be clear, if the objective is achieved, if the hypotheses are met, the interpretation of what has been provided, the limitations of the study and its way of solving it, etc.

Response:Thanks for the reviewer’s kind comment. We revised the discussion in the first paragraph. The Strengths and Limitations section were also revised, regarding to the way of solving methods, we provided in the “5. Conclusions and Future Works.” (page 12, line 358)

  • Indicators or indices on treatment fidelity should be provided (e.g., previously written protocol and if it was validated or this study is its validation; training of applicators; follow-up of its application; how it was guaranteed that it was applied to all participants in the same way...)

Response:Thank you so much for this important point. The fidelity part, we described in “2.8. Intervention Fidelity” and we have made minor modifications. (page 5, line 168)

  • The design of the intervention must be clear: number of groups; pre-post-monitoring measures; Session logs and analysis

Response:Thank you very much for this important point. We revised as followed.

  • A quasi-experimental, nonrandomized design, pre-post-monitoring measures was used. (page 2, line 94)
  • The MBSR intervention comprised two arm with 6 weeks of 2-h sessions held once a week. (page 4, line 153)
  • They were also required to record their practice time, post-practice awareness, stressful situations and stress responses, etc. on a construct paper log. (page 4, line 165)
  • The section of analysis, we stated in “2.10. Statistical Methods.” (page 5, line 188)
  • Data and evidence of the construct validity of the measures, convergent validity (extracted mean variance), discriminant validity, composite reliability or MacDonalds omega must be provided. If additional observations or tasks were included, how their inter-observer agreement was checked, how they were recorded, and how their accuracy was ensured

ResponseThank you very much for this important point. Our study did not use self-reported questionnaires, mainly to explore the application of objective instruments in breast cancer patients. In present study that we cited two studies to support our device has good reliability, a wristband heart rate monitor that is based on the photoplethysmogram (PPG) was used for continuous heart rate measurement. The measured data were collected and transmitted to the smartphone via Bluetooth 4.0. PPG instantly measures heart rate by estimating blood volume changes in blood vessels [27-28]. We calibrated the instrument prior to each measurement. (page 3, line 132)

[Reference in our manuscript]

  1. Chen, Y.C., et al., Artificial neural networks-based classification of emotions using wristband heart rate monitor data. Medicine (Baltimore), 2019. 98(33): p. e16863.
  2. Tsai, J.-C., et al., Design and Implementation of an Internet of Healthcare Things System for Respiratory Diseases. Wireless Personal Communications, 2021. 117(2): p. 337-353.

During the experiment, we did some qualitative interviews, in order to observe the changes and feelings of the breast cancer survivors. For example the following: (We anonymize the patient's statement)

I forgot to do mindful breathing at the beginning…I thought a lot at the beginning and would zone out during the process, but I was being able to absorb the lessons later. Mindful breathing helps to gradually slow my respiratory rate. (Sandy)

I currently have to work and undergo regular targeted therapy, so I don’t have much time to practice, but I will try mindful breathing and body scanning when I have the time. In the beginning, I couldn’t sense my body, but through rehabilitation…(Jane)

I don’t practice often because I’m too busy…I either have work to do or just want to sleep when I have time to spare. It is quite difficult for me to practice mindfulness in my daily life, but I can take time to practice it for a few minutes each day. (Mary)

I’m very impressed with the mindfulness program because I initially thought that this course was just a yoga course; however, I gained a lot by learning various mindfulness skills after joining the course. (Alanna)

During my hospitalization, I often felt very irritable, but after I performed mindfulness body scans and adjusted my breathing before going to bed, I found that the distracting thoughts in my mind slowly subsided. (Danielle)

  • All changes must be marked in color for easy checking

Response:Thank you very much. All text revisions have been marked in red in the revised manuscript. In the following are your comments (shown italics) and our detailed response (shown bold).

Authors are strongly encouraged to make the indicated changes in order to be recommended for publication.

Reviewer 2 Report

Thank you for your effort with this manuscript, I don't have further suggestions. Best regards

Author Response

Thank you for your effort with this manuscript, I don't have further suggestions. Best regards

Response:We sincerely appreciate your suggestions for our manuscript, thank you very much.

Round 2

Reviewer 1 Report

The paper is ready for publising

This manuscript is a resubmission of an earlier submission. The following is a list of the peer review reports and author responses from that submission.

Round 1

Reviewer 1 Report

The authors evaluated the physiological parameters of HRV under a mindfulness-based stress reduction program. They used two kinds of physiological measures to reduce stress to compare 272 the effectiveness of MBSR and UC. They also emphasized the 270 challenge of implementing such interventions in the context of stressful activities of breast 271 cancer patients. The topic in this paper is very interesting. However, there are some problems in the paper.

(1) The data collection and preprocessing processes need to be further elaborated.

(2) The authors used the standard deviation of normal-to-normal (SDNN) as the evaluation criterion, please explain in detail the mathematical meaning of this criterion and its significance from a medical perspective.

(3) Data on rectal cancer and lung cancer were introduced into this work to overcome the lack of statistical power. The authors need to show the correlation or similarity between the data distributions for these two diseases and breast cancer.

(4) The authors are encouraged to disclose their research data under the potential permissions, which can increase the impact of this work.

(5) Proofreading is required. some grammatical errors and inappropriate phrases are found.

Reviewer 2 Report

The study is valuable since it is an intervention that helps in the treatment of breast cancer patients, improving autonomous nervous system function.  

I understand the difference of the study is the measurement of HRV and other ANS parameters in those breast cancer patients after MBSR intervention. However, since this method was already used in other studies (involving participants with other characteristics, no breast cancer patients), I don’t find any novelty in this study. I believe the study could be of interest for the IJERPH readers because this intervention could be an important complementary treatment for cancer and other diseases’ patients.

The authors clearly state the strengths and limitations of their study, as well as future work. However, I have some minor concerns.

How the authors determined the ascertainment of exposure of participants? By self-report, secure records, structured interview, etc.?

Because of the small sample, the study has no representativeness

It could be interesting to compare (correlate) the physiological parameters measured with the self-reports of the patients.  

Reviewer 3 Report

It is an interesting and current intervention. However, it has important limitations that must be overcome and resolved before accepting your publication:

1) The empirical background that justifies the current study must conclude with the research question, specified in the objective and materialized in the hypotheses

2) A table with the intervention must be provided (sessions, phases, time dedicated to each one, procedure followed, aspects on which it focuses, participants, actions of the therapists, actions of the patients...)

3) Precise information must be provided about the intervention in the control group, because otherwise we cannot know if the efficacy is from the treatment or simply from receiving more attention for a specific time, and we know that it always improves, even if it is a nonspecific treatment. The fact that there is no more than one intervention group and with a single intervention modality is a serious problem that affects the generalisability of the results, and this situation must be clarified.

4) Follow-up data must be provided or at least included in the limitations of the study

5) A brief section should be included on the data analysis followed, its logic, the decisions made in this regard and why, and the software used

6) It is not enough to provide statistical significance, it is about providing practical significance, and for this, what is required is to provide the effect sizes in all cases, so they must be calculated (easy to do), provide them and interpret its practical or real significance (high, low, null...).

7) Since there are many modalities of mindfulness, what exactly was done should be clarified with more precision, including, in addition to the table of the complete intervention, another table with a type-session followed, how it was done, what procedures, what strategies , what content, how long, with how many participants each time...

8) Regarding treatment fidelity indicators, indications should be provided of: therapist training, treatment follow-up, prior written protocol for each session, guarantees of applying it to all participants in an equivalent manner... And the limitations in fidelity indicators should be indicated. this fidelity: there is no other comparable intervention group, nothing was done with the control group, no follow-up was done, samples from other centers were not used to increase their representativeness...

9) The references and sources must be updated using the most recent empirical studies (2022, 2021...), and including their evidence and arguments both in the background and in the discussion and conclusions. In the conclusions, the findings, interpretations, applications, limitations... must be related to the previous studies reviewed in the background, and in line with the justification made in the introduction.

10) All changes must be marked in color in the new version of the article, to facilitate verification. In addition, the authors must change all the indicated aspects, and respond one by one, when they send the new version of the article.

Authors are recommended to make the substantive changes requested, as soon as possible, and to re-upload the article, as instructed, in order to recommend its publication.

Reviewer 4 Report

Dear authors, you can find the suggestions for your manuscript.
